# Hydrogen Gas Improves the Postharvest Quality of Lanzhou Lily (*Lilium davidii* var. *unicolor*) Bulbs

**DOI:** 10.3390/plants12040946

**Published:** 2023-02-19

**Authors:** Hongsheng Zhang, Xuetong Wu, Xingjuan Liu, Yandong Yao, Zesheng Liu, Lijuan Wei, Xuemei Hou, Rong Gao, Yihua Li, Chunlei Wang, Weibiao Liao

**Affiliations:** 1College of Horticulture, Gansu Agricultural University, 1 Yinmen Village, Anning District, Lanzhou 730070, China; 2College of Life Sciences and Technology, Ningxia Polytechnic, 2 Xixia District, Yinchuan 750021, China

**Keywords:** fumigation, nutritional quality, color variation, browning degree, volatile compounds

## Abstract

Hydrogen gas (H_2_) is an important molecular messenger in animal and plant cells and is involved in various aspects of plant processes, including root organogenesis induction, stress tolerance and postharvest senescence. This study investigated the effect of H_2_ fumigation on the quality of Lanzhou lily scales. The results indicated the H_2_ remarkably declined the color variation and browning degree in Lanzhou lily scales by suppressing the activity of phenylalanine ammonia-lyase (PAL), peroxidase (POD) and polyphenol oxidase (PPO). Moreover, H_2_ significantly alleviated the degradation of soluble proteins and soluble sugars in Lanzhou lily scales during postharvest storage, mitigating the decline in nutritional quality. This alleviating effect of H_2_ might be achieved by increasing the endogenous H_2_ concentration. Collectively, our data provide new insights into the postharvest quality reduction of Lanzhou lily scales mitigated by H_2_ fumigation.

## 1. Introduction

Hydrogen gas (H_2_) is a colorless, odorless and tasteless gas, and is the structurally simplest gas in nature. In animals, H_2_ has been at the forefront of research on therapeutic medical gases [1]. Recently, H_2_ has been identified as a broad-spectrum anti-stress molecule that play an important role in plants. Hot spots of research on H_2_ have focused on response to heat stress [2], salt stress [3], UV irradiation [4] and cadmium stress [5]. In biological studies, besides inhalation of H_2_ [6], hydrogen-rich water (HRW) is considered a safe and easy method to mimic the physiological functions of endogenous H_2_ in plants and animals [7]. In addition, the H_2_ fumigation method is available in biological experimental studies. Recently, a study has shown that HRW application effectively prolonged the storage life in daylily buds (*Hemerocallis fulva* L.) [8]. Furthermore, H_2_ might maintain high nutritional quality and extend storage life of plants, such as kiwifruit [9], Chinese chives [10], cut rose and lily [8,11,12], litchi [13], lisianthus [14], pak choi [10]. In addition, the results of our team explored that the vase life of cut lily and cut rose was also improved in the presence of H_2_, and attributes these alleviative effects to the regulation of water balance and membrane stability by H_2_ [11,12]. Although it has been established that H_2_ plays an active role in postharvest preservation, it has yet to be clarified whether there are beneficial effects of H_2_ fumigation on the quality of postharvest bulb scales.

Lily is a perennial herb that is one of the most important bulb plants in the floral industry. There are three major edible lily species in China, Yixing lily, Longya lily and Lanzhou lily. Lanzhou lily (*Lilium davidii* var. *unicolor*), a variety of *L*. *davidii Duchartre*, is a perennial herb belonging to the genus *Lilium* in the lily family and is the “only sweet lily” in China [15]. The core cultivation area of Lanzhou lily is the mountainous regions of Gansu Province in Northwest China [16]. The special geographical conditions and climate make Lanzhou lily have unique qualities. It is rich in nutrients such as amino acids, proteins, starch, pectin and polysaccharides, as well as a variety of medicinal ingredients such as lilium saponins, flavonoids and colchicine [17,18,19]. Lanzhou lily can be used both as medicine and food. However, its postharvest storage and transportation faces problems with nutrient damage of fruits and vegetables. The fragile epidermis and high-water content in Lanzhou lily bulbs led to browning, rotting and spoilage during storage. Browning can be divided into two main categories, enzymatic browning and non-enzymatic browning, according to the mechanism by which it occurs. Enzymatic browning occurs in fresh fruit and vegetables when the tissue is bruised, cut and peeled and exposed to air and the polyphenols are catalyzed by polyphenol oxidase to o-quinone, which is further oxidized and polymerized to form brown pigments. The main enzyme closely associated with enzymatic browning is polyphenol oxidase (PPO), which is a copper ion-containing membrane protease. It mainly induces browning of tissues, produces off-flavors and causes loss of nutrients. Currently, low temperature is the main storage method during the postharvest of Lanzhou lily bulbs. In addition, some preservatives and chemical regulators are also used to keep lilies fresh during storage and transportation. Nonetheless, low temperatures trigger oxidative damage, which further results in nutrient changes in lilies [20]. Preservatives are potentially harmful to human health. Therefore, there is necessary to explore suitable and non-polluting techniques to reduce nutrient damage of Lanzhou lily during storage and transportation. Recently, gas fumigation has become the most popular alternative treatment for postharvest. Nitric oxide fumigation is effective in improving postharvest quality of strawberries [21]. Moreover, there is no literature on the effect of H_2_ on the postharvest quality of lily bulb scales in Lanzhou. Thus, the aim of this study was to explore the effect of H_2_ fumigation technology on the quality and scale color change of fresh lily bulb scales post harvesting at room temperature, especially its role in browning. These obtained findings have theoretical and practical implications for improving and maintaining the nutritional quality in postharvest Lanzhou lily scales, and provide a scientific basis and technical support for the transport and consumption of other perishable vegetables as well.

## 2. Results

### 2.1. H_2_ Fumigation Inhibited Color Variation and Browning Degree in Lily Scales

The browning degree of lily scales on the 3rd d in the control was 13.38, while the browning degree of lily scales with H_2_ fumigation was below 12 (Figure 1A). Furthermore, the browning degree of lily scales on the 6th d in the control was 18.12, while the browning degree of lily scales was extremely significantly reduced to 14.9 with H_2_ fumigation treatment. The browning degree of lily scales treated with H_2_ also decreased at the 9th, 12th and 15th d. In addition, Lanzhou lily scale epidermis color gradually browned from bright clean white to purplish red as the storage period prolonged (Figure 1B). The degree of browning in Lanzhou lily scales was significantly alleviated with H_2_ fumigation. Lily scales in the control began to brown on day 3, whereas H_2_-fumigated lily scales only began to brown slightly on day 6. The occurrence of browning was significantly delayed under the H_2_ fumigation.

The L* value of Lanzhou lily scales showed a downward trend during storage, and with H_2_ fumigation treatment, the L* value was higher than in the control at each time point (Figure 1C). There was a significant increase in the L* value of H_2_ fumigation-treated lily scales compared to the control. Unlike the L* value, the a* value increased gradually with storage time (Figure 1D). Notably, the reduction in a* value of lily scales at the 3rd, 6th, 12th and 15th d was significantly affected by the H_2_ fumigation treatment compared to the control. Additionally, H_2_ fumigation also increased the b* value of lily scales compared to the control, especially on days 3 and 6, when H_2_ fumigation treatment remarkably increased the b* value of lily scales, by 28.6% and 25.2%, respectively.

### 2.2. H_2_ Fumigation Enhanced the Activity of Three Browning-Related Enzymes

The PAL activity in Lanzhou lily scales showed a gradual upward with treatment time, as shown in Figure 2A. On days 3 and 6, there was an obviously significant difference in the PAL activity in control and H_2_ fumigation. Moreover, PAL activity was significantly reduced in H_2_ fumigation treatment compared to the control at 9, 12 and 15 d. The activity of POD in the Lanzhou lily scales first increased and then decreased slowly (Figure 2B). The POD activity was significantly decreased in H_2_-fumigated lily scales compared to the control. In particular, at the 3rd, 6th and 9th d, the POD activity was extremely significantly inhibited, by 79.7%, 66.4% and 59.4%, respectively. The trend of PPO activity changes was almost identical to that of POD (Figure 2C). Compared with the control, H_2_ fumigation remarkably decreased the PPO activity on days 3 and 6. The PPO activity on day 15 in H_2_-fumigated lily scales reached a maximum of 0.65 U g^−1^ FW, but it was still lower than that in the control scales. Therefore, H_2_ fumigation might alter the browning degree by downregulating PAL, POD and PPO activity in lily scales.

### 2.3. H_2_ Fumigation Increased the Nutrient Content in Lily Scales

The soluble protein content of Lanzhou lily scales increased slightly with prolonged storage time, with a peak on the 3rd d, followed by gradual decrease (Figure 3A). Compared to the control, the soluble protein degradation of Lanzhou lily scales with H_2_ fumigation was obviously alleviated, especially at the 3rd d and 6th d. Additionally, the soluble sugar content increased first and then decreased during the experiment (Figure 3B). Nevertheless, the H_2_-fumigated lily scales maintained a slightly higher soluble sugar content during storage than the control scales. Moreover, soluble sugar content was significantly increased by H_2_ fumigation on day 15. The total phenol content in Lanzhou lily scales showed an overall upward trend with increasing treatment time (Figure 3C). Higher total phenol content in H_2_ fumigation-treated scales was detected than control. Only on 6th day was the total phenol content of H_2_-fumigated lily scales significantly lower than in the control. Similarly, the flavonoid level in Lanzhou lily scales showed an upward trend (Figure 3D). Compared to untreated scales, the flavonoid level in H_2_-fumigated scales was remarkably decreased, by approximately 27.2% and 28.7% at the 9th and 12th d, respectively. In addition, anthocyanin content in Lanzhou lily scales first increased, then decreased gradually (Figure 3E,F). The anthocyanin content of the control scales reached a maximum of 1.3778 nmol·g^−1^ FW on day 12 and then began to decline. Compared to the control, anthocyanin content in Lanzhou lily scales treated with H_2_ fumigation was extremely reduced at each time point. In particular, the anthocyanin content of H_2_-fumigated lily scales was decreased by 81.7% and 86.5% at 6th and 12th d, respectively.

### 2.4. Principal Component Analysis in Lily Scales by H_2_ Fumigation

Principal component analysis (PCA) was performed on the dataset of the response values of E-nose to assess the effect of different storage times on the grouping of lily scales. The two-dimensional biplots of the score and loading of the lily samples are presented in Figure 4A. PC1 and PC2 represented 99.3 and 0.6% of the total variance, respectively, with the cumulative contribution rate of the first two PCs accounting for 99.9%, which indicated that they were sufficient to explain the total variance in the dataset. Compared with the control, H_2_ fumigation was separated from all treatments on the 3rd d, indicating that the aroma components of H_2_ fumigation on the 3rd d were significantly different from the control. Moreover, lily scales with H_2_ treatment greatly produced a volatile substance on 3 d. At other times, PCA plot in H_2_ fumigation probably overlapped, or was close to, that in the control, indicating that the aroma composition was basically similar, and flavor was not much different in H_2_ treatment and the control (Figure 4A). Additionally, it was observed that W5S, W3S and W2W were significantly positively correlated with the PC1 and were the main sensors for distinguishing different samples. The two lily samples for each treatment time were so closely related that they could be grouped into a small category at a Euclidean distance of 0.45 (Figure 4B). The lily scales (H_2_ fumigation and control) on day 3 belonged to one group and the rest of the samples to another group. The results of PCA and HCA in distinguishing the aroma of lily scales in H_2_ treatment and the control were completely consistent.

The radar fingerprint chart of the volatile compounds for the control and H_2_ fumigation was shown in Figure 4C. The chart depicted the responses of the ten sensor arrays to the volatile compound aroma collected during storage. The ten sensors gave similar signals for the volatile compounds in lily scales by illustrating the discrimination capability of each array. The response value distances of W5S, W2W and W1C were the largest while the distances of W6S, W5C, W1S, W1W, W2S and W3S were the smallest. Undeniably, W3C and W3S also largely contributed to the flavor discrimination of lily scales for different periods. Moreover, the radar fingerprint chart of volatile compounds aroma almost overlapped in “Day 0”, “Day 6”, “Day 9”,“Day 12” and “Day 15”, which indicated that similar volatile ingredients existed in these five time periods.

### 2.5. H_2_ Fumigation Increased the Endogenous H_2_ Content in Lily Scales

The endogenous H_2_ content in H_2_-fumigated lily scales basically first showed a slight increase followed by a gradual decrease during the storage period (Figure 5A). However, endogenous H_2_ content progressively decreased in the control. The endogenous H_2_ content of H_2_-fumigated lily scales reached a peak of 298.76 mol.kg^−1^ on the 3rd d. Additionally, compared to the control, endogenous H_2_ content was also significantly enhanced by H_2_ fumigation at 6, 9 and 12 d by 283.2%, 94.5% and 118.8%, respectively.

## 3. Discussion

The Lanzhou lily is renowned for its sweet flesh and rich nutritional value and is regarded as the “only sweet lily”. However, the color of lily scales was easily changed and turned browning in storage period [8]. The color change seriously affects the postharvest appearance quality of Lanzhou lily scales. In this study, it was found that the degree of color change in the scales increased with storage time. L* values on the scales surface decreased with increase time. The scales surface gradually changed from white to black. Moreover, the a* value tended to increase during storage. Recently, the gasotransmitter molecule has positive biological effectiveness on plant growth and development, environmental stress response and postharvest freshness preservation [22]. In exception to fumigating with different H_2_ concentrations, H_2_ is often dissolved in water to form HRW for immersion treatments to preserve horticultural products. In this study, results might suggest that H_2_ fumigation significantly alleviated the decrease in L* values while inhibiting the increase in a* values, thereby modulating color variation in Lanzhou lily scales. Similarly, incorporation of H_2_ into the packaging atmosphere decreased L* values and inhibited the increase in a* values in strawberries [23]. HRW treatment effectively suppressed surface yellowing of the fresh-cut Chinese water chestnut by modulating L*and b* value changes [24]. The color variation of lily was mainly violet-red and brown, and this study further examined the browning degree of the lily scales. H_2_ fumigation significantly inhibited the browning degree of lily scales. In addition, the results showed that lily scales with H_2_ fumigation obviously suppressed the PAL, POD and PPO activity compared to the control. This is consistent with the study of Li et al. [24] who reported that HRW could significantly decrease the PAL activity and the accumulation of flavonoids in fresh-cut Chinese water chestnut. PPO and POD were important factors leading to enzymatic browning in postharvest fruit and vegetables [25]. PPO is an important enzyme for the oxidation of phenolic compounds in enzymatic browning, which catalyzes the transformation from monophenols to o-diphenols and subsequently to o-quinones. The o-quinones transferred into brown, red and black pigments, which attributed to the browning [26]. Phenolic compounds also catalyzed oxidation by POD, as hydrogen peroxide existed [25]. Furthermore, HRW treatment delayed the pericarp browning and maintained the total soluble solids (TSS) of litchi fruit [13]. They also indicated that HWR treatment delayed pericarp browning of litchi, probably by inhibiting PPO and CAT activity. UV-C treatment was found to inhibit the browning index by about 37% [27]. They also found that the PPO and POD activities with UV-C treated lily scales were minimally reduced by about 26% and 18%, respectively. Moreover, 0.05 mM MT treatment remarkably declined the PPO and POD activity in fresh-cut taros [28]. Additionally, this study also found that the PPO, PAL and POD activities of H_2_-fumigated lily scales were significantly inhibited at the 3rd and 6th d. This is consistent with the time points of change in L* value, a* value, b* value and browning degree. Thus, the present study suggest that H_2_ fumigation might alter color variations and browning degree by downregulating PAL, POD and PPO activity in lily scales. Interesting, the present study also showed that H_2_ fumigation induced a significant endogenous H_2_ increase in Lanzhou lily scales during storage, especially on days 3 and 6 (Figure 5). During that process, the PPO, PAL and POD activity in H_2_-fumigated lily scales were significantly inhibited at 3rd and 6th d. Interesting, the color (L*, a* and b*) of H_2_-fumigated lily scales changed remarkably on the 3rd and 6th d. Even more, the onset of browning in lily scales fumigated with H_2_ was delayed until the 6th d. However, Lanzhou lily scales in the control began to brown on day 3. This could potentially reveal an important and close relationship between endogenous H_2_ and postharvest browning in lily scales. It is hypothesized that exogenous H_2_ fumigation might decrease the PAL, POD and PPO activity in lily scales by increasing endogenous H_2_ content, thus alleviating the browning and color variation of Lanzhou lily scales. Our research was the first to suggest that endogenous H_2_ production could be modulated by H_2_ fumigation to preserve postharvest Lanzhou lily scales.

Lanzhou lily scales are considered a “sweet lily” due to containing starch, sucrose, amino acids (polypeptides, proteins), fat, cellulose, glycosides, lipids and a variety of bioactive substances [29]. Postharvest of Lanzhou lily scales may lead to a large loss of nutritional quality and a decrease of edible quality [15]. Thus, here, we investigated whether H_2_ fumigation could maintain the nutritional quality in postharvest Lanzhou lily scales. Our results showed that H_2_ fumigation alleviated the decrease in soluble sugar and soluble protein content and suppressed the increase in total phenolic, flavonoid and anthocyanin content during storage at room temperature. Similarly, H_2_ (1%, 2% and 3%) application significantly prolonged the shelf life of chive leaves by abolishing the degradation of soluble proteins [10]. Furthermore, incorporation of hydrogen into the packaging atmosphere obviously exhibited higher total soluble solids [23]. However, they found that total phenolic content of the strawberry samples was significantly higher than the control, by 1.4 and 1.5 times. Jiang et al. [10] also demonstrated that total phenolic and flavonoid content in chive leaves was remarkably enhanced in the presence of H_2_. HRW, blue light + HRW and UV-B light + HRW treatments increased the content of total phenol and anthocyanin in immature radish microgreens, compared to blue light or UV-A irradiation alone [24]. Furthermore, incorporation of H_2_ into the packaging atmosphere further reduced the anthocyanin damage [23]. Moreover, the total phenol, flavonoid and anthocyanin contents in the HRW treatment were significantly higher than in the control litchi pericarp during the postharvest storage period [13]. However, in our study, H_2_ fumigation suppressed the increase of the total phenolic, flavonoid and anthocyanin content in lily scales. The possible reason for this difference is due to the different signaling pathways of secondary metabolites’ synthesis and degradation in different species, possibly due to differences in the way H_2_ is applied. Roubelakis-Angelakis and Kliewer proposed PAL activity as a factor that could influence the synthesis and accumulation of anthocyanins in grape berries. They also reported that PAL activity was positively correlated with the accumulation of anthocyanins. In our study, it also appeared to have a higher PAL activity in the untreated scales, which had higher anthocyanin content. Here, H_2_ fumigation treatment significantly inhibited the PAL activity in lily scales, thereby suppressing the increase in anthocyanin content. In addition, HRW treatment significantly reduced the accumulation of flavonoids in fresh-cut Chinese water chestnut [24]. Similarly, the increase in flavonoid content was also significantly inhibited by H_2_ fumigation in this study. Interesting, NaHS (0.8 mM), the donor of H_2_S, significantly reduced the accumulation of total phenols, anthocyanins and flavonoids in lily scales [8,15]. PAL is a key enzyme in flavonoid synthesis. In the study, H_2_ significantly inhibited PAL activity in lily scales, thereby suppressing the increase in flavonoid content. H_2_ might inhibit the increase in flavonoid content by inhibiting PAL activity. Here, H_2_ might be used as a preservative in the postharvest preservation of lily scales. Hence, H_2_ as a preservative might reduce nutrient loss during agricultural product storage.

In this study, H_2_ fumigation induced a significant increase in endogenous H_2_ in Lanzhou lily scales during storage (Figure 5). The result might reveal a close relationship between endogenous H_2_ level and postharvest quality in Lanzhou lily scales during storage. Similarly, exposure to HRW significantly alleviated the inhibition of root elongation and enhanced endogenous H_2_ concentration production in alfalfa seedlings [30]. Furthermore, H_2_-treated kiwifruit maintained a higher H_2_ concentration compared to the control in the storage period [9]. They indicated a possible relationship between H_2_ homeostasis and ripening in postharvest kiwifruit. Pre-harvest application of HRW to daylily buds mitigated the decline in endogenous H_2_ levels to a certain extent [8]. According to the principal component analysis diagram, W5S (responding to nitrogen oxides), W2W (responding to aromatic components and organic sulfur compounds) and W3S (responding to long-chain alkanes) sensors have obvious responses to aromatic components and content. Moreover, the aroma composition of H_2_-fumigated lily scales has a significant difference with control on the 3rd d. In addition, the endogenous H_2_ content in H_2_-fumigated scales peaked on the 3rd d in our study, suggesting that the 3rd d is a critical time point for lily scale preservation, which is consistent with the results of PCA and HCA. Intriguingly, the time points of changes in endogenous H_2_ content coincide with the time points of changes in PAL, PPO and POD activity and the onset of scale browning in lily scales. Thus, it is suggested that endogenous H_2_ might be involved in the onset of the browning of postharvest scales in Lanzhou lily during storage process.

## 4. Materials and Methods

### 4.1. Plant Materials and Experimental Treatments

The fresh bulbs used in this study were obtained from the Xiguoyuan of Qilihe District, Lanzhou, China. The bulbs were vacuum-packed with food-grade materials before storage and transported fresh using preservation cabinets at a storage temperature of 0–2 °C. Single head, consistent size, mechanical damage-free, pest-free and disease-free and white appearance of healthy Lanzhou lily bulbs were selected. The outer two layers of the bulbs and the other inner layers were discarded. The scales of the third, fourth and fifth layers were approximately the same size and similar thickness as experimental materials. The lily scales were rinsed three times with tap water, sterilized with 5% sodium hypochlorite for 15 min, finally washed with distilled water three times and placed on clean filter paper to dry naturally at room temperature. Dried scales (500 g) were placed in the fumigation unit (the fumigation bottle has inlet and outlet ports of air on both sides, with valves to control opening and closing). The H_2_ generator (QL-300, Saikesaisi Hydrogen Energy Co., Ltd., Shandong, China) was connected to the air inlet with a hose and continuously pumped in the hydrogen for 10 min. H_2_ fumigation in Lanzhou lily scales was continued for 3 d, which was renewed every 12 h. Fumigation bottles in the control group were filled with air. After fumigation, the lily scales (50 g) were randomly weighed and placed in a preservation box at room temperature (23 ± 2 °C) for 15 d. The sample was taken and photographed every three days to determine the relevant indicators. Each experiment was conducted in three biological replicates.

### 4.2. Assays for the Color Change

The surface color change of scales was estimated using a colorimeter (Model CR-400, Minolta, Tokyo, Japan), which showed the L*, a* and b* values by CIE color. The colorimeter was calibrated using a standard white tile (L* = 97.40, a* = −0.91, b* = 1.53).

The browning index (BI), which represents the purity of brown color (Palou et al., 1999), was calculated according to the following equation. Fresh lily scale as reference scale was shown in Figure A1.
$$\mathrm{BI}=\frac{100(x-0.31)}{0.172},\;\mathrm{where}\;x=\frac{a+1.75L}{5.645L+a-3.012b}$$

### 4.3. Assay of Phenylalamine Ammonialyase (PAL), Peroxidase (POD) and Polyphenol Oxidase (PPO) Activity

Lily scale (1 g) was ground with 5 mL phosphate buffer solution (PBS) (0.01 M, pH 7.4) on ice, rinsed with PBS (4 mL) and collected into a test tube. The supernatant was collected as the crude enzyme solution after centrifugation at 4000× *g* for 15 min at 4 °C. PAL activity was determined by measuring the absorbance of trans-Cinnamic acid converted from L-phenylalanine. A mixture of 3 mL PBS (50 mM, pH 8.8) and 0.5 mL L-phenylalanine (20 mM) was incubated at 37 °C for 10 min. After supernatant (0.5 mL) was added and mixed, the absorbance at 290 nm was quickly recorded as the original value (OD_0_). After that, the mixture was incubated at 37 °C for 60 min and stopped with 0.1 mL 6 M HCl. The OD value at 290 nm was measured again as the termination value (OD_1_). One U of PAL activity was defined as the amount of scale sample that caused a change of 0.01 in absorbance at 290 nm per minute. PAL activity was expressed in U g^−1^ FW.

Lily scales (0.5 g) were homogenized on ice with 5 mL of exaction buffer, consisting of 1 M PEG, 4% PVPP, 1% Triton X-100 and 0.1 M phosphate buffer (pH 5.5), and centrifuged at 12,000× *g* for 20 min. The supernatants were collected and used as enzymatic extracts. The POD activity was determined using a reaction mixture consisting of 25 mM guaiacol solution (3.0 mL), crude enzyme extract (0.5 mL) and H_2_O_2_ (0.2 mL). The POD activity was measured by recording the change in absorbance at 470 nm. One U of POD activity was defined as the amount of scale sample that causes a change of 0.01 in absorbance at 420 nm per minute. POD activity was expressed in U g^−1^ FW.

Lily scales (0.5 g) were homogenized in 5 mL of 0.1 M PBS (pH 6.0) containing 4% (*w*/*v*) polyvinylpyrrolidone, with the homogenate being centrifuged at 12,000× *g* at 4 °C for 15 min. The assay mixture contained 3.0 mL of 0.1 M PBS, 1 mL of 50 mM catechol and 0.1 mL of extract. The absorbance of three technical replicates was measured at 420 nm. One U of PPO activity was defined as the amount of scale sample that causes an increase of 0.001 in absorbance at 420 nm per minute. PPO activity was expressed in U g^−1^ FW.

### 4.4. Assays for Soluble Protein, Soluble Sugar, Total Phenolic, Total Flavonoid and Total Anthocyanin Content

Scales (0.5 g) were ground and extracted in 0.5 mM phosphate buffer (pH 6.0). The homogenate was centrifuged at 12,000× *g* for 20 min. Then, the supernatant (1 mL) and Coomassie brilliant blue (5 mL) were mixed. The absorbance at 595 nm was recorded after 2 min according to the method described by Bradford et al. [31]. The soluble proteins were expressed as milligrams per gram of FW.

According to the method of Yemm and Willis [32], lily scales (0.5 g) were ground into a homogenate, then distilled water (5 mL) was added and transferred to a graduated test tube. The test tubes were sealed with plastic wrap, boiled in boiling water and extracted for 30 min. The homogenate was cooled at room temperature and centrifuged for 10 min. The supernatant was collected into a volumetric flask (50 mL). The residue was re-extracted twice as described above and the supernatant was collected in the same volumetric flask. The reaction solution consisted of extracting solution (0.5 mL), 1.5 mL of distilled water, 0.5 mL of anthrone-ethyl acetate reagent and 5.0 mL of H_2_SO_4_ (98%). The absorbance was recorded at 630 nm at room temperature.

Total phenolic content was measured according to [33], with a minor modification. First, lily scale samples (0.5 g) were homogenized in liquid nitrogen and extracted by 50% ethanol (10 mL) and incubated at 70 °C for 1 h. The mixture was centrifuged at 12,000× *g* for 10 min and collected supernatant. Then, 1 M Folin–Ciocalteau reagent (1 mL) was added to the supernatant (1 mL) and incubated with 7.5% (*w*/*v*) Na_2_CO_3_ solution for 0.5 h at 30 °C in dark. The absorbance of the mixture was recorded at 765 nm. Total phenolic content was calculated from a standard curve for gallic acid and expressed as milligrams of gallic acid per g of FW of sample.

The total flavonoid content was determined by following the previous methods [33,34]. Scales (0.5 g) were ground in pre-cooled 1% HCL-methanol solution using a mortar and pestle and transferred into 10 mL tube. The homogenate was incubated for 20 min at 4 °C in dark and centrifuged (10,000× *g*) for 20 min. The supernatant (1 mL) was collected and mixed with 5% NaNO_2_ (0.3 mL). After 5 min, 0.3 mL of 10% Al (NO_3_)_3_ was added; then, after 6 min, 4 mL of 4% NaOH was added to the mixture; finally, distilled water was replenished to a constant volume of 10 mL. The absorbance at 510 nm was read and the content was expressed as milligrams per g of FW.

Total anthocyanins were extracted with a methanol-HCl method according to An et al. [10]. Briefly, samples (0.5 g) were soaked and incubated in 10 mL of 1% (*v*/*v*) methanol-HCl solution overnight in the dark at room temperature by the shaking bed. The absorbances were measured at 530, 620 and 650 nm using a spectrophotometer (UV-1800, Shimadzu, Kyoto, Japan). The total anthocyanin content was quantified using the following formula: OD = (OD_530_ − OD_620_) − 0.1 (OD_650_ − OD_620_). The total anthocyanin content was expressed as nmol per g of FW.

### 4.5. Assays for Electronic Nose

The headspace analysis was performed with a commercial PEN3.5 E-nose (Airsense Analytics, GmBH, Schwerin, Germany). The system contained 10 metal oxide sensors (namely, W1C, W5S, W3C, W6S, W5C, W1S, W1W, W2S, W2W and W3S). Prior to detection, each sample (2 g of scale samples) was placed in an airtight glass vial and closely capped with a PTFE–silicone stopper. Then, the samples were kept at 25 ± 1 °C for approximately 40 min (headspace generation time). The detection time of the sample was 120 s, the cleaning time of the sensor was 300 s and the adjustment time of automatic zero was 5 s. All samples were run with three repetitions.

The values of 101~110 s for each measurement using an E-nose were imported into WinMuster software and repeated 3 times to generate a principal component analysis (PCA) figure. PCA employs the idea of dimensionality reduction to simplify problems. A plurality of number indexes interconnected to each other were translated into several comprehensive and unrelated indicators, which are the principal components of the original multiple indexes. The between-group linkage method with a metric of Euclidean distance was performed to apply hierarchical cluster analysis (HCA) in this study. The merged data presented as a dendrogram, where the horizontal axis represented the Euclidean distance amongst groups and the vertical axis indicated the lily scale flavor similarity. The data obtained in Winmuster were averaged in excel to calculate the response values of the ten electronic metal sensors for the control and H_2_ fumigation during the storage period, and radar plots were generated using the data analysis tool.

### 4.6. Assays for Endogenous H_2_ Concentrations

Concentrations of endogenous H_2_ in scales were determined by gas chromatography. Briefly, samples (0.5 g) were homogenized with distilled water (7 mL) for 3 min. The homogenate was transferred to a brown bottle and octanol (5 μL) and H_2_SO_4_ (2.5 mM) were added. Then, pure nitrogen gas was bubbled in a vial to displace the air. Afterwards, the vial was immediately capped and shaken vigorously by hand for approximately 2 min. The brown flask was heated at 70 °C for 1 h to liberate H_2_ from the lily scales. When cooled at room temperature, gas sample (100 μL) was collected in the headspace of the brown bottle with a gastight syringe and immediately injected into an Agilent 7890 gas chromatograph (Gas chromatography 7890, Agilent Technologies Inc., Palo Alto, CA, USA) for analysis of hydrogen concentrations of three technical replications.

### 4.7. Statistical Analysis

The results were expressed as the means ± standard error (SE) from three independent experiments with three biological replicates for each. Microsoft Excel 2016 and SPSS 22.0 software (SPSS Inc., Chicago, IL, USA) were used to analyse data. The independent *t*-test (*p* < 0.05 or *p* < 0.01) was used to analyse the significance differences for treatments on the same day.

## 5. Conclusions

In summary, the results suggest that H_2_ significantly alleviated the postharvest quality decline in Lanzhou lily scales. H_2_ fumigation remarkably reduced the color variation and browning degree in Lanzhou lily scales by inhibiting the activity of PAL, POD and PPO. Moreover, H_2_ significantly maintained the nutrient content in lily scales during postharvest storage. In addition, H_2_ fumigation induced the production of endogenous H_2_, suggesting that endogenous H_2_ is involved in the process of lily quality maintenance. Hence, we provide strong evidence for the involvement of H_2_ in the postharvest quality maintenance of lily scales. Although this study contributes to the understanding of the mechanisms by which H_2_ fumigation might improve the quality of postharvest lily scales, the molecular mechanisms are still unclear. Therefore, future studies on H_2_-regulated bulb postharvest quality should focus on gasotransmitter signaling molecular interactions, protein modifications and transcript abundance during storage.

## Figures and Tables

**Figure 1 plants-12-00946-f001:**
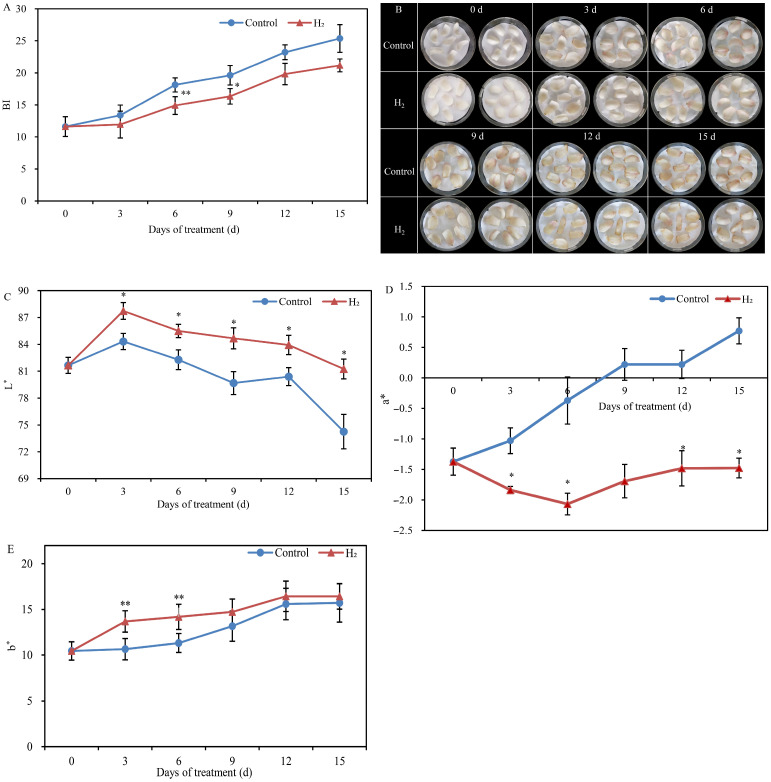
Changes in the browning index (BI) (**A**), phenotypic (**B**), L* value (**C**), a* value (**D**) and b* value (**E**) of Lanzhou lily scales fumigated with H_2_ at room temperature. The error bars indicate the means ± SE (n = 15 scales for each of three independent experiments). Values with asterisks are significantly different (* *p* < 0.05; ** *p* < 0.01) from the control treatment on the same day according to independent *t*-test. d: days.

**Figure 2 plants-12-00946-f002:**
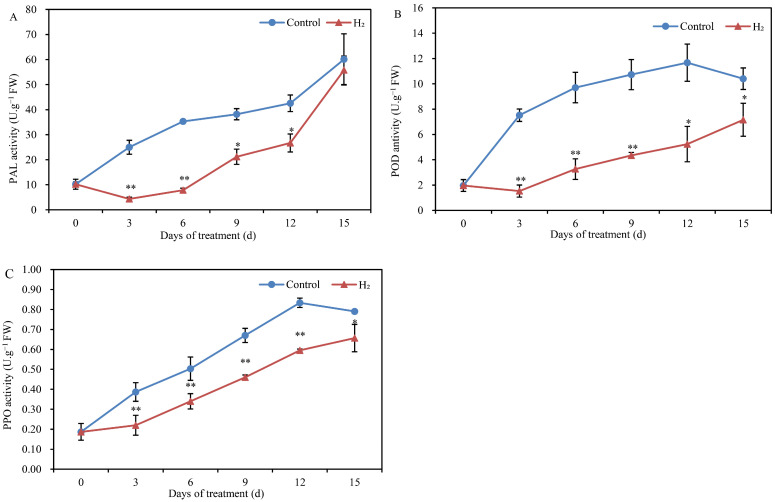
Changes in enzyme activities in Lanzhou lily scales with or without H2 fumigation during storage at room temperature for 15 days. (**A**) PAL, phenylalanine ammonia-lyase; (**B**) POD, peroxidase; (**C**) PPO, polyphenol oxidase. The error bars indicate the means ± SE (n = 9 scales from each of three independent experiments). Values with asterisks are significantly different (* *p* < 0.05; ** *p* < 0.01) from the control treatment on the same day according to independent *t*- test. d: days.

**Figure 3 plants-12-00946-f003:**
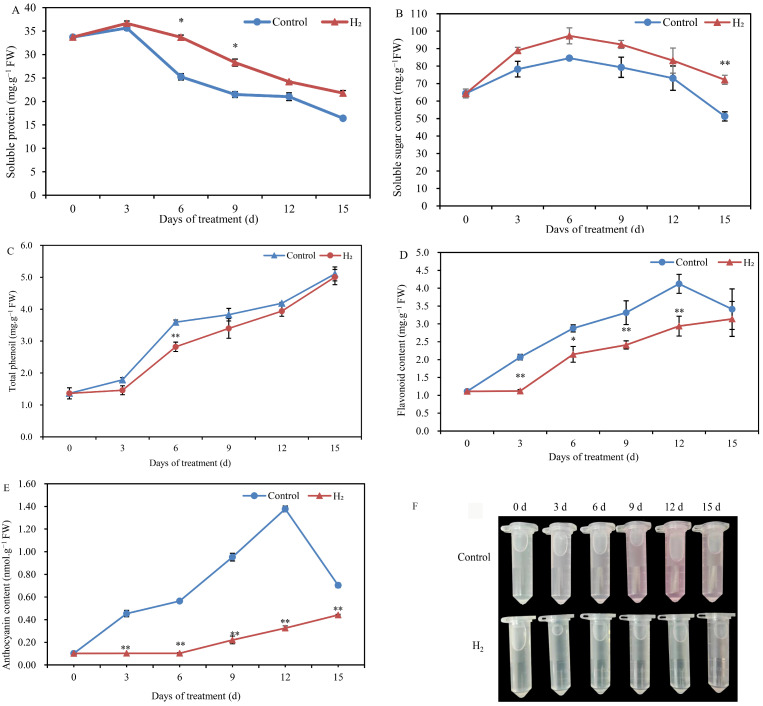
Changes in the contents of soluble protein (**A**), soluble sugars (**B**), total phenols (**C**), flavonoids (**D**), anthocyanidin (**E**) and anthocyanin extract solution color (**F**) of Lanzhou lily scales fumigated with or without H_2_ during storage at room temperature for 15 days. The error bars indicate the means ± SE (n = 9 scales from each of three independent experiments). Values with asterisks are significantly different (* *p* < 0.05; ** *p* < 0.01) from the control treatment on the same day according to independent *t*-test. d: days.

**Figure 4 plants-12-00946-f004:**
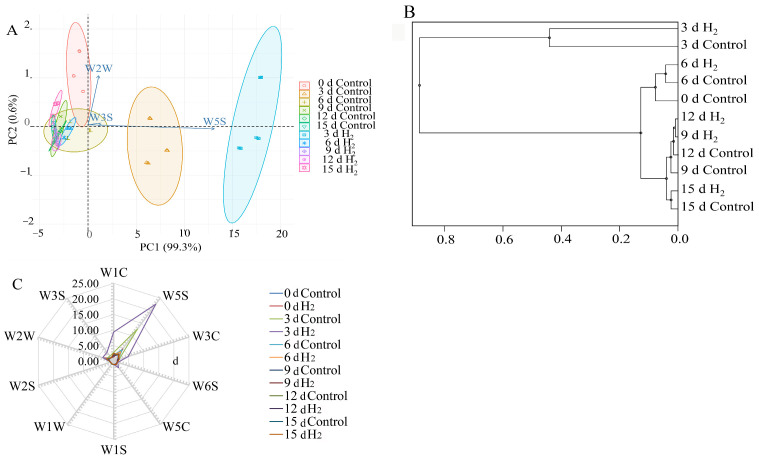
Principal component analysis (PCA) (**A**) and hierarchical cluster analysis (HCA) (**B**) of Lanzhou lily scales during storage based on E-nose date. (**C**): Radar fingerprint chart of the volatile compounds in Lanzhou lily scales during storage. W1C: Aromatic components, benzene; W5S: Nitrogen oxides; W3C: Ammonia, aromatic components; W6S: Hydrogen; W5C: Alkane aromatic components; W1S: Short-chain alkanes; W1W: Inorganic sulphones; W2S: Alcohols; W2W: Aromatic components, organic sulphones; W3S: Long-chain alkanes.

**Figure 5 plants-12-00946-f005:**
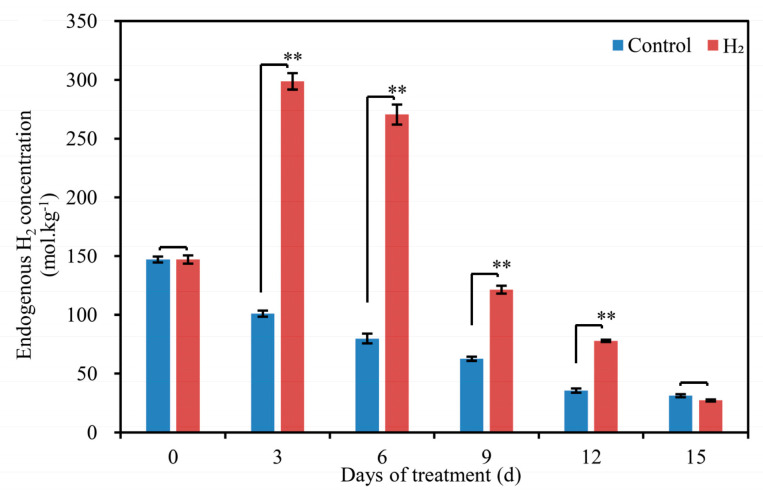
Changes in endogenous H_2_ content in Lanzhou lily scales with or without H_2_ fumigation during storage at room temperature for 15 days. The error bars indicate the means ± SE (n = 9 scales from each of three independent experiments). Values with asterisks are significantly different (* *p* < 0.05; ** *p* < 0.01) from the control treatment on the same day according to independent *t*- test. d: days.

## Data Availability

Not applicable.

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
