# Peer review of "Hydrogen Gas Improves the Postharvest Quality of Lanzhou Lily (*Lilium davidii* var. *unicolor*) Bulbs"

_plants, 2023, doi:10.3390/plants12040946_

Round 1

Reviewer 1 Report

Review 2052386:

The manuscript investigated the effects of H2 fumigation on the quality of Lanzhou lily scales. The qualities and browning-related enzymes of Lanzhou lily scales have been comprehensively analyzed. However, this manuscript should be performed some revisions before being accepted for publication.

1.     The keywords were not accurate enough.

2.     The abbreviations (Hydrogen gas) have appeared in the abstract.

3.     The mechanism of browning can be explained in the introduction.

4.     The format of some units was incorrect. Please check the full manuscript.

In addition, the language of this manuscript needs to be improved.

Reviewer 2 Report

This paper found that H2significantly declined browning and the decline in nutritional quality of Lanzhou lily scales during postharvest storage. The paper is clearly presented, however, some concerns should be addressed.

Abstract 

Which declines in nutritional indicators of Lanzhou lily scales are significantly mitigated by H2during post-harvest storage?

Figures

1. In Figure 1B, add a space between the number and d.

2. The X coordinate time (d) should be changed more specifically.

3. In Figure 3B, -1 superscript.

4. In Figure 4A and B, please enlarge the font of the words.

4. In Figure 4C, add a space between the number and d. please enlarge the font of the words. Note the writing specification of H2. The meanings of W1C, W2C, W3C… should be explained in the figure legend.

5. Line 283, Fig 4A should be Figure 4A.

6. Line 298, Fig 4C should be Figure 4C. Please check similar problems in the full text.

Materials and methods

Under what temperature conditions of Lanzhou lily scales were transported.

Discussion

Please discuss how the increase of endogenous hydrogen inhibits browning of Lanzhou lily scales.

Conclusion

Increase research deficiencies and prospects.

Reviewer 3 Report

REVIEW REPORT

“Hydrogen gas improves the postharvest quality of Lanzhou Lily (Lilium davidii var. unicolor) scales”

The investigation is focused in H2 treatments to mitigate postharvest effects of bulbs. Authors analyze physiological parameters of bulbs as color. Endogenous concentration of H2, soluble protein and sugar, and total content of phenols, flavonoids and anthocyanins are quantified too.

COMMENTS

-       In the title you have to add the word "bulb”

QUESTIONS

-       How the hydrogen gas application have an incidence in the flavour of Lanzhou lily bulbs?

-       What about the physiological parameters as weight and size of bulb across postharvest treatment days?

-       Is possible to apply H2 of another ways?

In general, the manuscript is good, but it is necessary to edit the document with more precision in paragraphs inside of methodology.

Reviewer 4 Report

Please consider the following suggestions:

1)      review punctuation and spacing in the whole manuscript;

2)      section 2.2, a reference scale is missing, i.e. the reader does not find indications about the maximum and minimum of the scale, what are conventionally the minimum performance values;

3)      section 2.4, use or FW o fresh weight, not a mix of them;

4)      section 2.5, unclear exposure. At times, the reader has the impression of reading the instruction manual of the instrument rather than its description;

5)      section 2, the Authors are advised to choose the past tense to describe the whole experiment, rather than the present tense as happens in some sections;

6)      section 3, the exposition of the results should follow the order of the variables as reported in the materials and methods;

7)      section 3.1, only the significative results should be reported, and not significative ones should only be hinted at (browning degree section);

8)      section 3.2, H2 Fumigation Enhanced the Activity of Three Browning-Related Enzymes: the corresponding section in the materials and methods is missing;

9)      line 254-256 and 320-321,  this phrase seems inappropriate for the results, it would be more appropriate in the discussion section suitably motivated;

10)  check the way in which the units of measurement are reported, and align them with those reported in the International System;

11)  section 3.4, in the materials and methods section the description of how the Authors performed the principal component analysis in this experiment is missing;

12)  line 268-271, it is strongly suggested that this sentence be deleted;

13)  line 285-291, the Authors should explain the use of a statistical test, the modality and the relative motivation in the appropriately dedicated section of the materials and methods not in the results section;

14)  radar fingerprint, see point 11;

15)  section 3, before using an acronym in the text, Authors should report the full form and the acronym in brackets for the first time;

16)  line 343 ‘ Incorporation ‘;

17) section 4, the Authors assert that during the storage period the concentration of some compounds tends to increase. However, the dry matter content at the various storage dates is not shown. the authors should show some evidence that this is not a water concentration phenomenon due to tissue loss of water, rather metabolic processes. The Authors are invited to investigate the metabolic mechanisms behind the results obtained in relation to the investigated variables, and not only to report similar experiments existing in the literature that can support or refute their data. 

Round 2

Reviewer 2 Report

1. Line 36, missing full stop after [1].

2. Figure 1, please place the pictures in order of A, B, C, D, E. Figure 1B, 6d with a blank space for number and letter. 

3. Please keep the font size of A, B, C… the same in all figures. The font sizes in the figures are also inconsistent, e.g. a* in Figure 1D and b* in Figure 1E.

4. Figure 3 should preferably be placed on the same page, -1 superscript in the Figure 3C.

5. Please check the units in figure 5, and the image resolution is too low.

6. Line300, 0.39U.g-1FW, Remove the ., 0.39U space in middle. Similar issues checked in the manuscript. 

7. The border thicknesses are also very inconsistent in the same figure, e.g. Fig. 1D and E.

Author Response

Comments 1:

  1. Line 36, missing full stop after [1].

Response:

Thank the reviewers for their careful reading of this manuscript. We have revised this manuscript. The related statement is as follow:

Line 29: “.” was added.

Comments 2:

  1. Figure 1, please place the pictures in order of A, B, C, D, E. Figure 1B, 6d with a blank space for number and letter.

Response:

We greatly thank you for your comment. We have replaced Figure 1 in order. Additionally, we have also made modifications to Figure 1B. Please see the revised Figure 1 for details.

Comments 3:

  1. Please keep the font size of A, B, C… the same in all figures. The font sizes in the figures are also inconsistent, e.g. a* in Figure 1D and b* in Figure 1E.

Response:

Thank you very much for your comment and we are very sorry for this error. We have standardized the font size in all figures. Please see the revised Figure for details.

Comments 4:

Figure 3 should preferably be placed on the same page, -1 superscript in the Figure 3C.

Response:

Thank you very much for your comment and we are very sorry for this error. In Figure 3C, “-1” has been revised superscript. Please see the revised Figure 3C for details.

Comments 5:

  1. Please check the units in figure 5, and the image resolution is too low.

Response:

We greatly thank you for your comment. We have replaced the Figure 5. Please see the revised Figure 5 for details.

Comments 6:

  1. Line300, 0.39U.g-1FW, Remove the ., 0.39U space in middle. Similar issues checked in the manuscript.

Response:

We greatly thank you for your comment. We have revised the manuscript. Please see the revised manuscript for other similar revisions.

Comments 7:

  1. The border thicknesses are also very inconsistent in the same figure, e.g. Fig. 1D and E.

Response:

Thank you very much for your comment and we are very sorry for this error. We have standardized the border thicknesses of Figures. Please see the revised Figure for details.

Reviewer 4 Report

The manuscript investigated the effects of hydrogen fumigation on the postharvest quality of Lanzhou lily scales. The manuscript was revised, improved and the topic has been comprehensively analyzed. 

Author Response

Thank the reviewers for their careful and patient reading and review of our manuscript.

Thanks again for your constructive comments, which are very helpful for us to improve the quality of the manuscript.